# Therapeutic Efficacy of Ultraviolet C Light on Fungal Keratitis—In Vitro and Ex Vivo Studies

**DOI:** 10.3390/antibiotics14040361

**Published:** 2025-04-01

**Authors:** Mark A. Bosman, Jennifer P. Craig, Simon Swift, Simon J. Dean, Sanjay Marasini

**Affiliations:** 1Department of Ophthalmology, The University of Auckland, Auckland 1142, New Zealandjp.craig@auckland.ac.nz (J.P.C.);; 2Department of Molecular Medicine and Pathology, The University of Auckland, Auckland 1142, New Zealand; s.swift@auckland.ac.nz

**Keywords:** microbial keratitis, ultraviolet C light, *Candida*, *Aspergillus*, corneal ulcer, light-based anti-infective technology

## Abstract

**Objective:** Fungal corneal infections are challenging to treat due to delayed diagnostic procedures, bacterial co-infections, and limited antifungal efficacy. This study investigates the therapeutic potential of ultraviolet C (UVC) light alone and combined with antifungal drugs. **Methods:** A subsurface infection model was developed in semi-solid agar droplets, with *Candida albicans* cells or *Aspergillus brasiliensis* spores inoculated into 0.75% *w/v* yeast peptone dextrose (YPD) agar in a 96-well microplate (5 µL per well). Two treatment groups were tested: (1) UVC exposure (265 nm, 1.93 mW/cm^2^) for durations of 0 s, 5 s, 10 s, 15 s, 30 s, 60 s, or 120 s, and (2) UVC combined with antifungal drugs (Amphotericin B and Natamycin) at their minimum inhibitory concentrations (MICs), determined in YPD broth. After treatment, agar droplets were homogenized, diluted, and plated for microbial enumeration. The most effective UVC doses were further tested in an ex vivo *C. albicans* porcine keratitis model, where the corneal epithelium was debrided, infected with *C. albicans*, and exposed to UVC. Corneas were then homogenized and plated to evaluate treatment efficacy. **Results:** UVC exposure of ≥15 s inhibited *C. albicans,* and ≥10 s inhibited *A. brasiliensis* (all *p* < 0.05). The broth MICs were 0.1875 µg/mL for Amphotericin B against *C. albicans*, 6.25 µg/mL against *A. brasiliensis*, and 0.78125 µg/mL for Natamycin against *C. albicans*, 7.8125 µg/mL against *A. brasiliensis*. The broth MIC did not eradicate fungi in the subsurface model. Combined treatments enhanced inhibition (all *p* < 0.05), with 30 s UVC + amphotericin B for *C. albicans* (*p* = 0.0218) and 30 s UVC + natamycin for *A. brasiliensis* (*p* = 0.0017). Ex vivo, 15 s and 30 s UVC inhibited growth (*p* = 0.0476), but no differences were seen between groups (all *p* > 0.05). **Conclusion:** UVC demonstrated strong antifungal efficacy, with supplementary benefits from combining UVC with low doses of antifungal drugs.

## 1. Introduction

Microbial keratitis (MK) is a major, preventable cause of blindness and vision impairment worldwide [1]. Rapid, intensive treatment is required to manage the condition and prevent permanent vision loss [2]. Treatment of MK primarily involves the use of antifungal drugs [3], and, in many countries, patients are often left with long-lasting visual impairment due to the inaccessibility of expensive treatment routes [4]. The development of antifungal resistance by MK pathogens adds further difficulties [5,6,7]. For these reasons, it is increasingly attractive to establish therapeutic pathways that effectively treat a wide range of infections and are less prone to microbial resistance. Ultraviolet (UV) light has been used safely and effectively in the treatment of a wide range of corneal conditions and ectatic diseases such as keratoconus [8]. In the latter case, UV light (370 nm) is used to excite riboflavin, a chemical substrate, to induce a photodynamic reaction, in the procedure known as corneal collagen cross-linking [9,10]. This method not only treats corneal ectatic diseases but also microbial keratitis through the production of free oxygen radicals [11]. Previous work by our group has demonstrated the efficacy of treatments that use short wavelength (265 nm) UV light alone, classed as UVC, for treating bacterial infections in vitro and in vivo [12]. UVC’s antimicrobial action is attributed to its ready absorbance by microbial DNA, causing cross-linking of adjacent base pairs by photochemical reaction, ultimately resulting in cell death [13,14]. Therefore, no additional chemical substrate is required to inhibit microbial growth using this wavelength of light.

While the efficacy of UVC to treat bacterial corneal infections has been previously established in preclinical studies, many cases of MK are due to infections caused by fungal agents such as *Candida* and *Aspergillus* species [15,16,17]. Since differentiating the causative infective organism on clinical presentation is challenging and time-consuming, and the confirmatory laboratory diagnostic testing can result in delayed treatment, it would be valuable to establish whether UVC has broad-spectrum capabilities to be used as an empiric treatment.

Pathological fungi typically present as yeasts or molds and often require more intensive treatment than bacteria due to their ability to form hyphae and grow in deeper tissues [18]. To simulate a subsurface infection where fungi grow both on and beneath a surface, we created an in vitro model in which fungi were mixed with molten, nutrient-rich agar. As the mixture cooled to room temperature, it formed a semi-solid droplet, effectively representing a fungal corneal infection model with nutrient-rich medium. The treatment efficacy was tested in this model. Given UVC’s limited depth penetration through a medium [19], the efficacy of UVC in inhibiting fungal growth is currently unknown. Therefore, the study aimed to evaluate the antifungal efficacy of UVC against *Candida albicans* and *Aspergillus brasiliensis*, fungi associated with MK [15]. Additionally, the study investigates the potential synergistic effects of combining UVC with antifungal agents, natamycin, or amphotericin B.

## 2. Materials and Methods

### 2.1. Microbe Preparation

Suspensions of *C. albicans* were prepared by inoculating 10 mL of Difco yeast peptone dextrose broth (YPD, Fort Richard, Auckland, New Zealand) with 2–3 colonies of *C. albicans* ATCC 10231, acquired from the American Type Culture Collection (Manassas, VA, USA), and incubating overnight at 37 °C with shaking (200 rotations per minute, rpm). After 18–20 h, the suspension reached 10^8^ colony-forming units (CFU)/mL. The following day, the suspension was centrifuged at 2500 rpm for 10 min, the broth was replaced with sterile saline, and the process was repeated to resuspend *C. albicans* in fresh saline.

*A. brasiliensis* ATCC 16404, also acquired from the American Type Culture Collection, was cultured on Difco yeast peptone dextrose agar (Fort Richard, Auckland, New Zealand) for 2–3 weeks at 26 °C. Spores were collected by gently scraping them from the agar surface but without filtering into 10 mL sterile saline with an L-shaped spreader rod. The resulting suspension was pipetted into a 50 mL V-bottom tube and thoroughly vortexed, which obtained the saturation point of 10^7^ spores/mL when counted by a haemocytometer.

### 2.2. UVC Light Source

The light-emitting diode (LED) has a fixed intensity of 1.93 mW/cm^2^, projecting a 4.5 mm diameter spot at 8 mm distance [12]. The LED (SETI UVTOP265) emits at 264.5 nm with a 12.3 nm halfwidth (Hopoocolor OHSP-350UV, Hangzhou, China). At 8 mm distance, the beam has a homogeneous flat-top profile, verified by fluorescence photography and beam profiling in ImageJ (version 1.54), showing consistent exposure across the 4.5 mm diameter with rapid intensity fall-off at the edges. Exposure durations were controlled by an electric timer (dose = intensity × duration).

### 2.3. Preparation of In Vitro Fungal Keratitis Model

An in vitro infection model simulating subsurface corneal infections was adapted from previous work using *Pseudomonas aeruginosa* [12]. In brief, 100 µL of either *C. albicans* (10^7^ cells) or *A. brasiliensis* (10^6^ spores) was mixed with 1 mL of warm molten YPD top agar (0.75% *w/v* agar in YPD) and aliquoted into a 96-well flat-bottom microplate. Three 5 µL (containing 10^5^ cells of *C. albicans* or 10^4^ spores of *A. brasiliensis*) were added to the center of each well. Once solidified, the droplets formed a circular subsurface fungal corneal infection model, which was used to assess the therapeutic potential of UVC, both alone and combined with conventional antifungal drugs, in three independent experiments. The experimental protocol has been represented graphically in Figure 1.

### 2.4. Therapeutic Efficacy of UVC Alone

Agar droplets were exposed to UVC for 0 s (control), 5 s, 10 s, 15 s, 30 s, 60 s, or 120 s. After exposure, droplets were dissolved in 300 µL saline by crushing and vigorous pipetting, then transferred to 700 µL sterile saline for a final volume of 1 mL. For *C. albicans*, the mixture was diluted 1:10, and 10 µL was plated on YPD agar and incubated at 37 °C for 24 h for colony counting. For *A. brasiliensis*, the mixture was diluted 1:2, and 50 µL was plated on YPD agar and incubated at 26 °C. After 40–42 h of incubation, spores were counted individually based on their fringe or star-shaped appearance.

### 2.5. Therapeutic Efficacy of UVC Plus Antifungal Drugs

To assess the potential synergistic effect between UVC and traditional antifungal agents, an objective drug dosage standard was established at a sub-inhibitory concentration. The minimum inhibitory concentrations (MIC) [20] of amphotericin B and natamycin were determined as both are commonly used in treating fungal keratitis worldwide [17,21]. The MIC determined in YPD broth conditions (liquid medium) was used to establish the objective drug dosage for the combination therapy in the semi-solid in vitro infection model described above.

#### 2.5.1. MIC Assays of Antifungal Drugs

To determine MICs, both natamycin and amphotericin B were dissolved in DMSO. Natamycin was prepared at a stock concentration of 1 mg/mL and amphotericin B at 5 mg/mL in DMSO. These stock solutions were stored and subsequently diluted to working concentrations using serial dilution as needed. Three columns of a sterile 96-well plate were designated as technical repeats. The lower 7 rows of each column were filled with 50 µL of YPD broth. Then, 100 µL of either natamycin or amphotericin B was added to the top row. Serial dilutions were performed by transferring 50 µL from the top row to the row below, diluting the concentration at each step. The final row had a 50 µL volume after discarding the excess solution. A 50 µL sample of *C. albicans* or *A. brasiliensis* at 10^6^ CFU/mL was added to each well and incubated overnight in a sealed, humidified container. *C. albicans* was incubated at 37 °C at 200 rpm, while *A. brasiliensis* was incubated at 26 °C with shaking. After 24 h, optical density was measured at 600 nm using a VICTOR Nivo Multimode Plate Reader (PerkinElmer, Waltham, MA, USA) to determine the MIC, defined as the lowest dose that inhibited growth. Here, we took a reduction in turbidity of at least 90% (i.e., MIC_90_) to indicate growth inhibition.

#### 2.5.2. Antifungal Efficacy of Combined Treatment

The MICs of the drugs were determined to be as follows: 0.1875 µg/mL for amphotericin B against *C. albicans*, 6.25 µg/mL against *A. brasiliensis*, and 0.78125 µg/mL for Natamycin against *C. albicans*, 7.8125 µg/mL against *A. brasiliensis*. Once MIC doses of both drugs were established in liquid broth, the agar droplet technique was used to study the combined effects of antifungal drugs and UVC. Droplets were exposed to antifungal drugs only, 15 s or 30 s UVC, 15 s or 30 s UVC + drug, or no treatment (control). Natamycin (10 µL) and amphotericin B (100 µL) were applied at their MICs, with UVC delivered first for combined treatments. After one hour, droplets were homogenized in saline, serially diluted, and plated for microbial enumeration. In preliminary tests evaluating the antifungal efficacy of natamycin and amphotericin B at their respective MICs using the in vitro semi-solid infection model, 10 µL of each drug was applied to assess treatment effectiveness compared to an untreated control. The results indicated that while 10 µL of natamycin demonstrated the desired partial efficacy in inhibiting fungal growth at its MIC, a higher volume of amphotericin B was required to achieve similar effects. Consequently, the antifungal volumes were adjusted accordingly for each drug.

### 2.6. Assessment of UVC Effectiveness Against C. albicans Ex Vivo

As a step toward future testing for potential clinical translation in an in vivo model with active immunity, the goal of the current experiments was to explore the in vitro findings using an ex vivo porcine keratitis model. Corneal integrity was assessed visually to confirm that the epithelium was intact, and any eyes with epithelial damage were excluded. Intact corneas were dissected from tissue blocks, leaving a small radius (~2 mm) of limbal tissue, then stored in sterile phosphate-buffered saline (PBS). After washing with warm water for one minute, they were dipped in 5% povidone iodine, followed by three PBS washes, and stored in sterile PBS at 4 °C for up to one week. Then, the ex vivo porcine corneas were inoculated with *C. albicans* to evaluate UVC treatment efficacy in postmortem tissue sourced from a local slaughterhouse.

Processed corneas were placed on a petri dish with the epithelium facing up and debrided using an Alger Brush (The Alger Company Inc., Lago Vista, TX, USA) to create a 2 mm circular wound. A 10 µL saline solution containing 10^6^ CFU of *C. albicans* was applied to the wound, and the corneas were incubated at 37° C for 2 h. After three washes with sterile PBS, the corneas were exposed to one of the following treatments: 0 s (control), 15 s, or 30 s UVC. The treated region was excised using a 2.5 mm diameter biopsy punch, homogenized in 2 mL sterile saline using a Multi-Gen 7XL homogenizing package (Thomas Scientific, Swedesboro, NJ, USA), and 50 µL plated on YPD agar. After 24 h at 37 °C, CFU were counted. The experiments were performed in triplicate over three days.

### 2.7. Statistical Analyses

Data were confirmed for normality using the Shapiro–Wilk test in GraphPad Prism Version 9.3.1, and a one-way ANOVA with multiple comparisons was applied to compare treatment efficacy across the treatment groups. A threshold of *p* < 0.05 was considered statistically significant.

## 3. Results

### 3.1. Therapeutic Efficacy of UVC on Subsurface In Vitro Infection Model

Overall, ANOVA indicated a statistically significant difference between group means for *C. albicans* growth (F = 15.67, *p* < 0.0001). Holm–Šídák’s multiple comparisons identified specific significant differences in *C. albicans* CFU count after 30 s, 60 s, and 120 s, versus control (*p* = 0.0002, *p* < 0.0001, *p* < 0.0001, respectively). In each of these cases, a reduction in CFU count was seen, with a mean reduction of 14.08 CFU/10 µL at 30 s (95% CI: 11.51, 23.32). At 60 s, there was a mean reduction of 16.61 CFU/10 µL (95% CI: 9.62, 20.15), and at 120 s, the mean decrease was 23.17 CFU/10 µL (95% CI: 3.0, 13.66). Comparisons between UVC-exposed groups revealed significant differences between shorter and longer durations. Specifically, the 5 s exposure group showed significant differences from the 30 s, 60 s, and 120 s exposures (*p* = 0.0082, 0.0015, and <0.0001, respectively). Similarly, the 10 s exposure group differed from the 30 s, 60 s, and 120 s exposure groups (*p* = 0.0154, 0.0028, and <0.0001, respectively). The minimum UVC dose that resulted in a statistically significant growth inhibition of *C. albicans* when compared to the untreated control group was 15 s (*p* = 0.0169, highlighted in blue). Statistically significant differences from control were also demonstrated for 60 s (*p* = 0.0001) and 120 s (*p* = 0.0001) (Figure 2).

An analysis of group means to evaluate the dose-dependent UVC antifungal efficacy in the *A. brasiliensis* in vitro keratitis model revealed a statistically significant effect using ANOVA (F = 5.665, *p* = 0.0001). Holm–Šídák’s multiple comparisons indicated significant differences between the control group and all UVC-exposed groups with exposure durations of 10 s or longer (Figure 3A,B). At 10 s, the mean reduction was 36.43 spores/50 µL (*p* = 0.00062, 95% CI: 3.91, 68.95), and at 15 s, the reduction was 35.88 spores/50 µL (*p* = 0.0281, 95% CI: 2.413, 69.34) compared to the control. At 30 s, the mean reduction was 50.10 spores/50 µL (*p* = 0.0003, 95% CI: 17.58, 82.62), at 60 s, this difference was 38.00 spores/50 µL (*p* = 0.0165, 95% CI: 4.54, 71.46), and at 120 s, the reduction was 45.65 spores/50 µL (*p* = 0.0014, 95% CI: 13.13, 78.17). A comparison between exposure groups revealed a statistically significant difference between the 5 s and 30 s groups with a mean difference of 35.78 spores/50 µL (*p* = 0.0167, 95% CI: 4.23, 67.33) (Figure 3A,B). The minimum UVC dose that resulted in statistically significant growth inhibition of *A. brasiliensis* was 10 s (*p* = 0.0062, highlighted in blue).

### 3.2. Minimum Inhibitory Concentration Assays of Antifungal Drugs

After incubation with the microorganisms at 37 °C for 24 h, turbidity appeared in all wells at lower doses. Absorbance assays were also performed, which confirmed that these doses represented the minimum concentrations, as they did not show a significant difference from positive control. In contrast, doses below these minimum concentrations showed a significant difference from the positive control (*p* < 0.0001 in all cases) (Figure 4). All doses above the threshold also differed significantly from the negative control (*p* < 0.0001 in all cases). Based on these results, the MICs of natamycin for *C. albicans* and *A. brasiliensis* were determined to be 25.0 µg/mL and 62.5 µg/mL, respectively. Similarly, the MICs of amphotericin B for *C. albicans* and *A. brasiliensis* were 1.5 µg/mL and 50.0 µg/mL, respectively.

### 3.3. Therapeutic Efficacy of UVC Plus Antifungal Drugs in the In Vitro Fungal Keratitis Model

After determining the MICs of the antifungal drugs in broth conditions and the minimum inhibitory UVC doses in the in vitro models, the combination of UVC with amphotericin B on *C. albicans* and UVC with natamycin on *A. brasiliensis* were tested. The combination treatments of UVC and antifungal drugs led to statistically significant reductions in microorganism load for both species.

Exposure to UVC and amphotericin B significantly reduced *C. albicans* CFU counts (F = 0.674, *p* = 0.0218) (Figure 5A). Independent multiple comparisons demonstrated a mean reduction of 9.56 CFU/10 µL after 15 s plus amphotericin B (*p* = 0.0114, 95% CI: 6.47, 18.41) and 12.67 CFU/10 µL after 30 s UVC plus amphotericin B exposures (*p* = 0.0010, 95% CI: 4.95, 13.72) relative to untreated controls (Figure 5B).

Combined exposure to UVC and natamycin reduced mean spore counts of *A. brasiliensis* by 40.72 CFU/50 µL after 15 s UVC + natamycin (*p* = 0.0296, 95% CI: 25.74, 81.82) and 64.94 spores/50 µL after 30 s UVC + natamycin exposures (*p* = 0.0003, 95% CI: 13.53, 45.58). A significant reduction was also seen when comparing the natamycin + 30 s UVC group with the natamycin-only group (*p* = 0.0125, 95% CI: 7.76 to 82.02) (Figure 6A,B).

### 3.4. Efficacy of UVC Alone in C. albicans Ex Vivo Keratitis Model

The ex vivo study of UVC antifungal efficacy in porcine *C. albicans* keratitis (Figure 7) demonstrated a downward trend in CFU count at higher UVC doses and a significant overall difference in ANOVA (*p* = 0.0476). However, multiple comparisons confirmed no significant difference between groups at *p* < 0.05 confidence level. However, at the 90% confidence level, the difference was closer to statistical significance between control and both 15 s and 30 s exposures (*p* = 0.0577 in both cases).

## 4. Discussion

*Candida* and *Aspergillus* species are among the predominant species implicated in MK, a corneal infection, [22,23] and represent a strong starting point for understanding the efficacy of this therapy in the realm of fungal keratitis management. *C. albicans* and *A. brasiliensis* were tested as model pathogens in the present study. The data presented here strongly indicate an inhibitory effect of UVC exposure on both fungal species tested in an in vitro fungal corneal infection model. Efficacy was dependent on UVC exposure duration, with *A. brasiliensis* responding to exposures as low as 10 s and both species responsive to exposures of 15 s and longer. For both species, there was also no significant difference seen in outcomes in exposures beyond 30 s, indicating that any compounding effects of further treatment may not be sufficient to warrant risks imposed by excessive UVC exposure. In clinical practice, use of an empiric dosing strategy that demonstrates broad effectiveness against a wide range of pathogens is preferable to avoid reliance on awaiting test results to identify specific species, which may delay treatment onset. Thus, it would be valuable to compare 15 s UVC, which has been previously established preclinically to treat bacterial keratitis [12], with 30 s UVC in further in vivo studies to establish the optimal dose that offers a “broad spectrum” antimicrobial treatment.

These findings support and expand upon prior studies that have demonstrated a therapeutic effect of direct UVC exposure to a range of pathogens in numerous experimental settings [12,24,25,26]; however, this is the first study to examine the effect of UVC in a subsurface infection-like in vitro model. Previous testing with this model demonstrated an optimum dose at 15 s for bacteria, so it may be necessary to expand the dosing strategy when fungal infections are suspected or if seeking a “one-size-fits-all” treatment strategy. To this end, further safety testing may also be beneficial. In previous studies, safety assessments were performed based on 15 s treatment according to bacterial susceptibility, which suggested that in a single application or in a cumulative dose of 45 s over three days, such UVC exposure was safe in terms of DNA defects in the mouse model tested. These data highlight the likely safety of 30 s UVC but determining the upper safe limit of therapeutic UVC in managing corneal infections would be valuable if repeated exposures are required [19].

Pharmacological treatment remains the gold standard for MK pathology, and UVC treatment shows potential as an appealing adjunct therapy in clinical practice. Dual-therapy treatments can serve to limit the dose of individual components required for equal or superior combined efficacy, enhancing the safety profile of the overall treatment [27]. Thus, to this end, we aimed to study the effect of UVC in concert with drug treatment.

The MICs of each drug were established as a relevant baseline from which to study the efficacy of UVC in an in vitro environment, such that clinical doses of the drug did not eliminate the colonies of exposed microbes outright. Exposure to combined therapy methods in both fungal species showed significant effectiveness at both 15 s and 30 s, relative to control, for *A. brasiliensis* exposed to natamycin and for *C. albicans* exposed to amphotericin B. At a minimum, the study confirms that the method is effective in reducing fungal growth and does not adversely affect drug efficacy.

Furthermore, a significant reduction in spore formation was seen in the 30 s combined-treatment group with *A. brasiliensis* and natamycin, indicating that combined treatment was more effective than drug treatment alone in this species. While the same was not seen in the *Candida*-Amphotericin B group, it was observed that, where *Candida* samples were not responsive to 15 s of UVC alone, they became responsive to this exposure duration when also exposed to drugs in the combined-treatment study. This may indicate that, while combined therapy does not necessarily enhance the efficacy of the treatment outcome with this drug and microbe combination, it may allow for efficacy at a lower dose of UVC, which may reduce the risk associated with longer exposure periods, improving the overall safety profile, and allowing benefits from shorter exposures. The latter is particularly beneficial because shorter treatment is more practical in a clinical setting and is, therefore, more appealing to both clinicians and patients. Overall, these data suggest the value of further study involving combined treatments, especially using amphotericin B for the treatment of *C. albicans* infection and natamycin for *A. brasiliensis* infections.

The effect of UVC on tissue that had been inoculated ex vivo was studied to understand how outcomes might translate to a more clinically relevant setting than can be achieved in vitro. The availability of quality tissue and the methodological difficulties that followed presented challenges and limitations for applying the method broadly, and this meant that an equivalent protocol for *A. brasiliensis* could not be established in these experiments. However, with *C. albicans,* some evidence emerged to indicate the therapeutic effect of UVC in the ex vivo infection model. An overall trend toward reduced expression is seen in the data, with significance met at the 90% level (*p* = 0.0577). This is most likely due to the high degree of variability seen in the distribution of data points. Variability is not unusual with microbial samples due to the nature of living organisms, but the limitations imposed by processing ex vivo tissue may need to be countered by highly powered studies or refinements in the methodological techniques.

Managing yeast and fungal infections can be challenging due to the presence of slowly growing hyphae [28]. Consequently, the in vitro and ex vivo infection models used in these experiments may not fully represent the slowly progressing fungal infections that take time to show clinical signs. UVC light’s limited penetration through the cornea [19] highlights the benefit of combining it with antifungal medications, as supported by the current study’s data. These results suggest that UVC could be a promising empirical treatment for microbial keratitis, applied directly at the infection site after tissue debridement and sample collection for laboratory cultures [29]. Given that fungal corneal infections are typically not confirmed until later in their progression, early UVC application as soon as an infiltrate appears may improve treatment outcomes as an empirical approach, inhibiting microbial growth regardless of the infection’s origin. UVC’s limited corneal penetration also reduces off-target damage [30], and its efficacy against fungal pathogens can be enhanced through debridement and combination with antifungal drugs. Additionally, when compared to other phototherapeutic techniques, UVC acts without other agents, whereas other photodynamic procedures, e.g., corneal collagen cross-linking, require UVA and riboflavin for their antimicrobial effects [9,10].

In our previous research, we tested a broad range of microorganisms using agar plates as surface lawns without considering depth factors, except when evaluating *Pseudomonas aeruginosa*. In the current experiments, we specifically examined the depth factor by utilizing an in vitro infection model and an ex vivo model to study infection in subcellular contexts for fungal keratitis. These new data build upon our previous in vitro studies and lay the groundwork for transitioning to in vivo research.

The potential for UVC to induce sub-inhibitory dose-induced fungal tolerance is worthy of future research. While ecological evidence shows that some microorganisms can develop tolerance to UV light and other stressors over extended periods, a recent review found limited evidence for tolerance to sub-lethal doses of light-based anti-infective treatments [31]. However, considering ecological examples, it remains a possibility worth exploring, particularly in regard to how fungal species might respond to prolonged or repeated sub-lethal therapeutic UVC doses.

## 5. Conclusions

To conclude, UVC alone showed potent antifungal activity, but combined therapy of UVC and antifungal drugs was deemed more effective in managing *C. albicans* and *A. brasiliensis* infections. These data warrant further in vivo testing to explore the therapeutic potential of UVC as an empiric treatment for microbial keratitis.

## Figures and Tables

**Figure 1 antibiotics-14-00361-f001:**
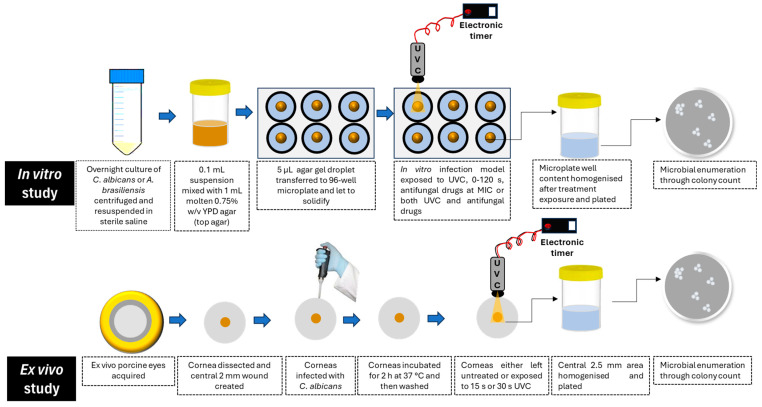
Graphical representation of the experimental protocol.

**Figure 2 antibiotics-14-00361-f002:**
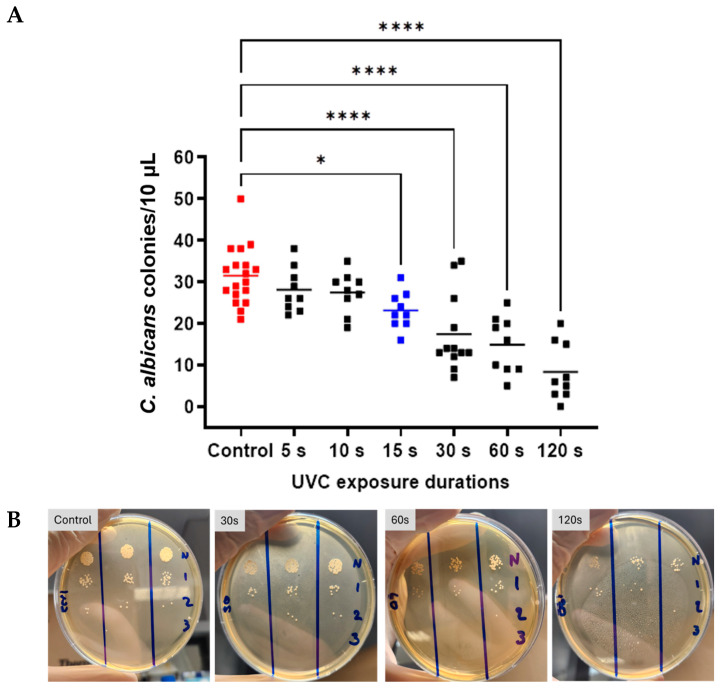
Antifungal efficacy of UVC in an in vitro *C. albicans* keratitis model. Graph (**A**) demonstrates CFU counts for control and UVC-exposed groups (5–120 s exposures). Individual data points (*n* = 9) are shown for all technical (*n* = 3) and biological repeats (*n* = 3), and horizontal bars represent group means. Representative plates (**B**) *C. albicans* CFU counts at increasing UVC doses. Dilution series identifies CFU counts at 1:10 dilution (N = neat sample, 1 = 10×, 2 = 100×, 3 = 1000×) intervals starting with undiluted sample. Significance was determined by ANOVA with multiple comparisons, * *p* < 0.05, **** *p* < 0.001.

**Figure 3 antibiotics-14-00361-f003:**
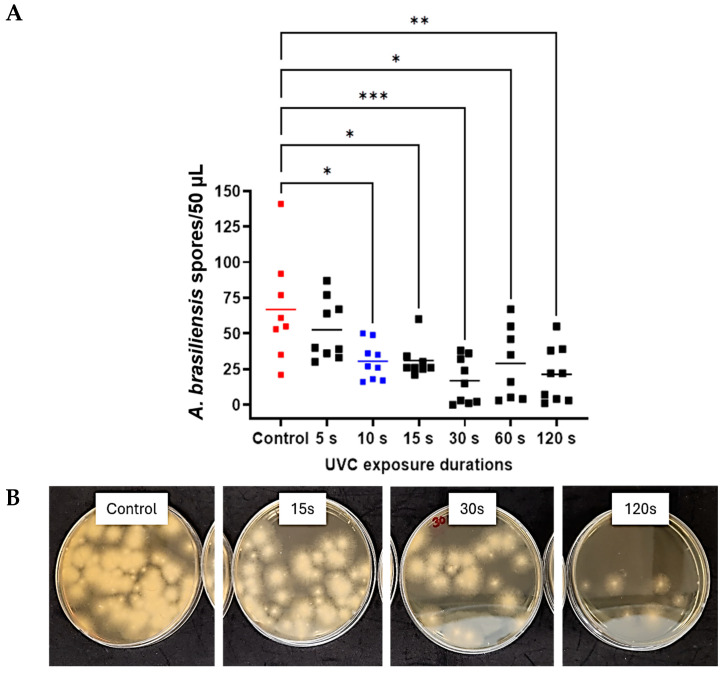
Antifungal efficacy of UVC against *A. brasiliensis* in vitro corneal infection model. Individual data points (*n* = 9) are shown for all technical (*n* = 3) and biological repeats (*n* = 3), and horizontal bars represent group means (**A**) Representative agar plates (**B**) Spore-forming units at increasing UVC doses. Significance was determined by ANOVA with multiple comparisons, * *p* < 0.05, ** *p* < 0.01, *** *p* < 0.001.

**Figure 4 antibiotics-14-00361-f004:**
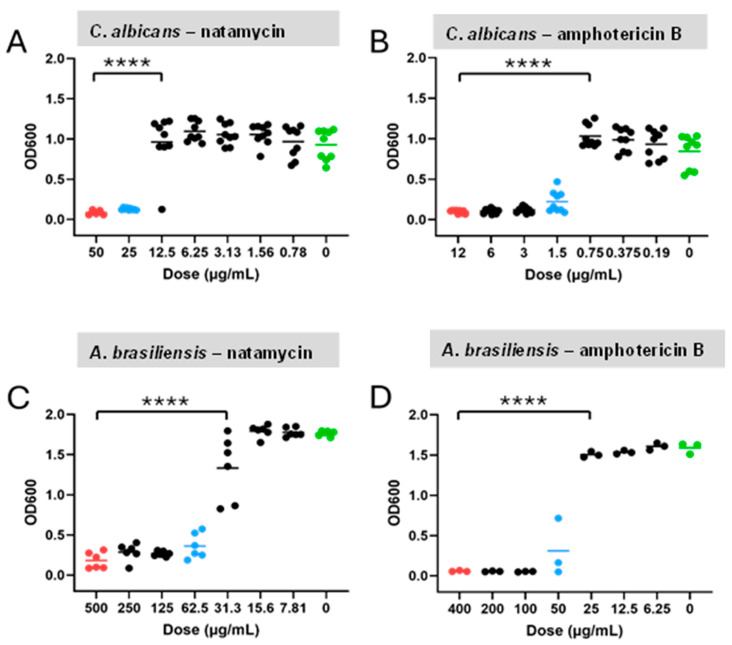
OD600 assays decreasing drug doses for *C. albicans* and *A. brasiliensis*. Individual data points and group means (horizontal bars) are shown for the *C. albicans*–natamycin (**A**), *C. albicans*–amphotericin B (**B**), *A. brasiliensis*–natamycin (**C**), and *A. brasiliensis*–amphotericin B (**D**). Red dots = positive control; blue dots = minimum inhibitory concentrations; and green dots = untreated negative control. Significance was determined by ANOVA with Tukey’s multiple comparisons (*p* < 0.0001), **** *p* < 0.0001.

**Figure 5 antibiotics-14-00361-f005:**
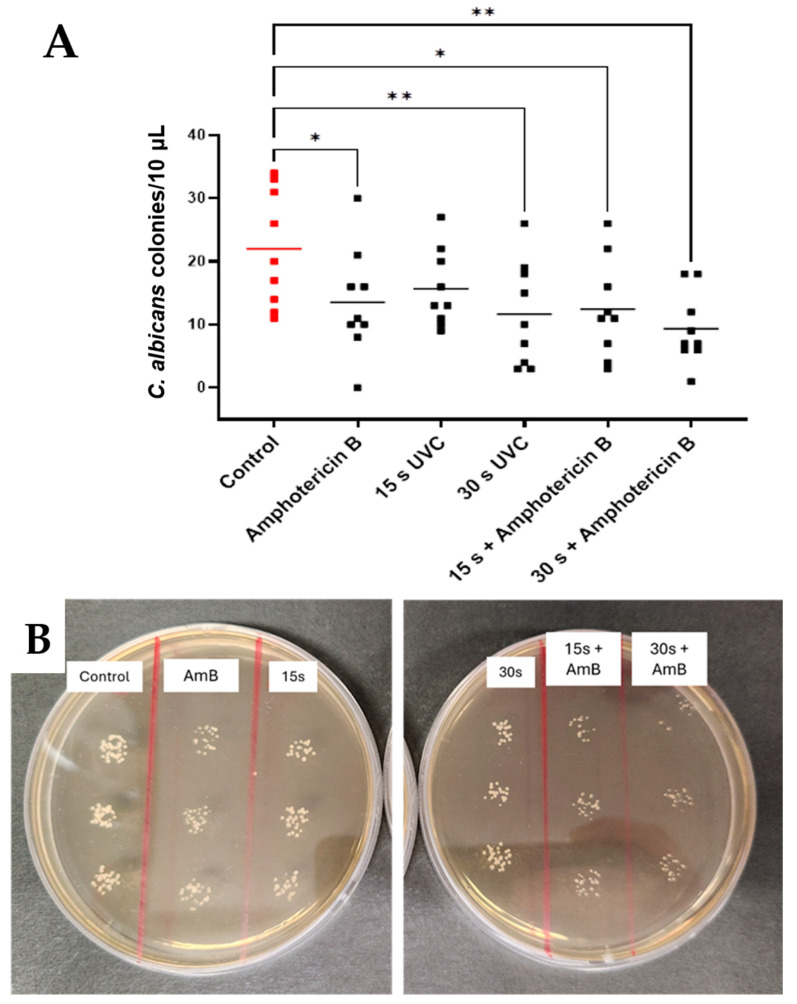
Effects of combination therapy using UVC plus amphotericin B on *C. albicans* in subsurface infection-like in vitro model. Individual data points (*n* = 9) are shown for all technical (*n* = 3) and biological repeats (*n* = 3), and horizontal bars represent group means (**A**). Representative plates (**B**). Visible differences in CFU counts across treatment groups (control, amphotericin B, and 15 s and 30 s UVC and combined therapies). Significance was determined by ANOVA with multiple comparisons, * *p* < 0.05, ** *p* < 0.01.

**Figure 6 antibiotics-14-00361-f006:**
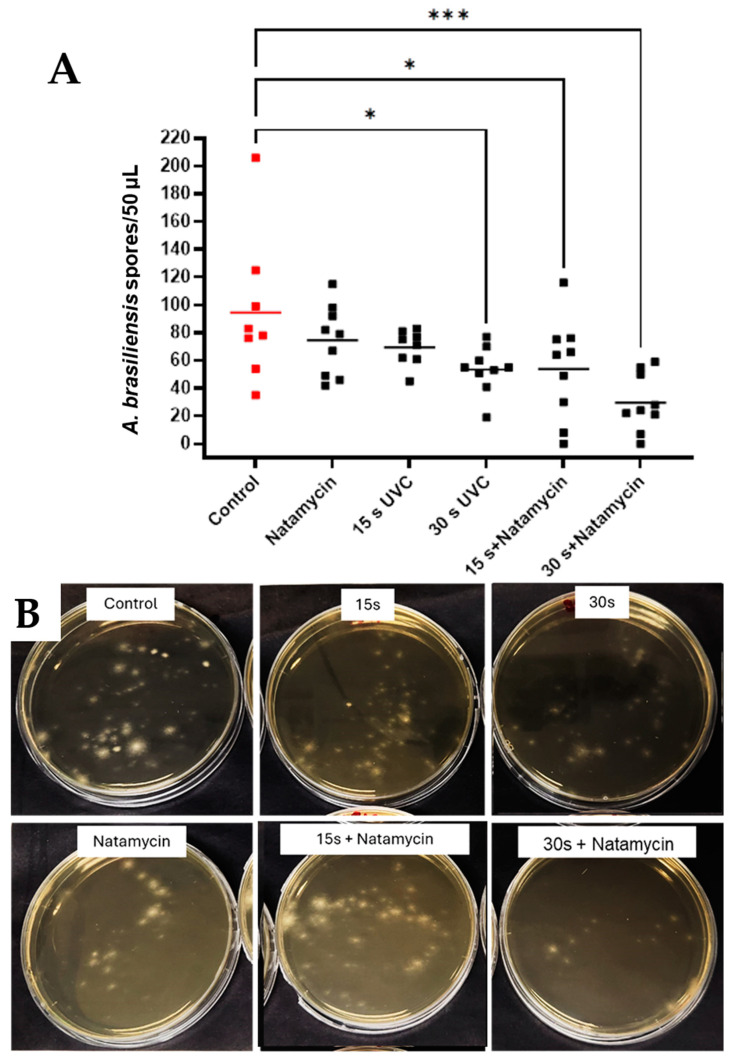
Effects of combination therapy using UVC and natamycin in *A. brasiliensis* in the subsurface infection-like in vitro model. Individual data points (*n* = 9) are shown for all technical (*n* = 3) and biological repeats (*n* = 3), and horizontal bars represent group means (**A**). Representative plates (**B**). Visible differences in spore quantification across treatment groups. Data were compared using ANOVA and Tukey’s multiple comparison test, * *p* < 0.05, *** *p* < 0.001.

**Figure 7 antibiotics-14-00361-f007:**
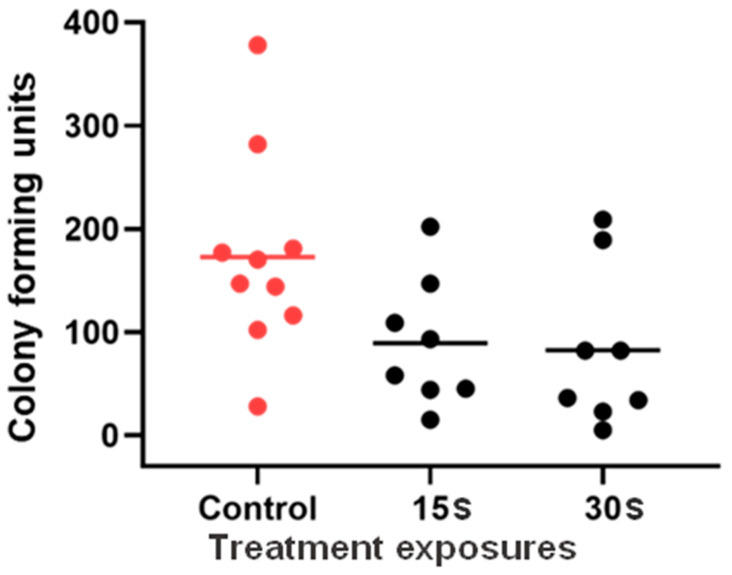
UVC efficacy in an ex vivo porcine *Candida albicans* keratitis model. Group means and individual data points are presented. Data were compared with ANOVA and Tukey’s multiple comparisons test (*n* = 8 to 10).

## Data Availability

The original contributions presented in this study are included in the article. Further inquiries can be directed to the corresponding author.

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
