# Peer review of "Therapeutic Efficacy of Ultraviolet C Light on Fungal Keratitis—In Vitro and Ex Vivo Studies"

_antibiotics, 2025, doi:10.3390/antibiotics14040361_

Round 1

Reviewer 1 Report

Comments and Suggestions for Authors

The study explores the therapeutic effects of UVC light, both alone and in combination with antifungal drugs, on fungal keratitis. It includes in vitro and ex vivo models, evaluating the inhibitory effects of UVC at different exposure durations on Candida and Aspergillus. The experimental design is well-structured, data analysis is thorough, and the conclusions hold certain clinical relevance. Overall, this study is innovative and has clinical application prospects, making it suitable for publication after revisions.
该研究探讨了UVC光单独使用和与抗真菌药物联合使用对真菌性角膜炎的治疗效果。它包括体外和离体模型,评估UVC在不同暴露时间对念珠菌和曲霉的抑制作用。实验设计合理,数据分析透彻,结论具有一定的临床意义。然而,有几个方面需要改进,以提高研究的严谨性和翻译潜力。总体而言,本研究具有创新性和临床应用前景,适合修订后发表。

    1. The authors repeatedly mention the limited penetration ability of UVC:
        "Given UVC’s limited depth penetration through a medium [20], it was hypothesized that UVC would not be effective in inhibiting fungal growth."
        "Since UVC light does not penetrate deeply through the cornea [20], combining UVC treatment with antifungal medications supports effectiveness, as indicated by the current study's data."
        "Debriding the corneal epithelium will also improve UVC penetration into the stroma, as the epithelium poses a barrier to UVC light reaching the anterior stroma [20]."
1.作者反复提到UVC有限的渗透能力:
“鉴于UVC通过介质的有限深度渗透[20],假设UVC不会有效抑制真菌生长。"
“由于UVC光线不会穿透角膜[20],因此UVC治疗与抗真菌药物相结合支持有效性,正如当前研究的数据所示。"
“清除角膜上皮也将改善UVC渗透到基质中,因为上皮对UVC光到达前基质构成屏障[20]。"

    Does this imply that UVC’s limited penetration is a fundamental technical limitation that is difficult to overcome? Compared to other phototherapy techniques, such as photodynamic therapy (PDT), what advantages does UVC offer?
这是否意味着UVC的有限渗透是一个难以克服的基本技术限制?与其他光疗技术(如光动力疗法(PDT))相比,UVC有哪些优势?

   2. Although the study cites previous literature on UVC-induced DNA damage, it does not directly assess the potential harm of UVC to corneal cells in this research.
2.尽管该研究引用了以前关于UVC诱导的DNA损伤的文献,但在本研究中并没有直接评估UVC对角膜细胞的潜在危害。

    3. The study does not evaluate whether prolonged or repeated use of UVC could lead to sub-inhibitory dose-induced fungal tolerance.

3.该研究没有评估是否长期或重复使用UVC可能导致亚抑制剂量诱导的真菌耐受性。

Comments on the Quality of English Language

The English language quality of this paper is generally good, with the rigor and readability of an academic paper.

Author Response

General comment: The study explores the therapeutic effects of UVC light, both alone and in combination with antifungal drugs, on fungal keratitis. It includes in vitro and ex vivo models, evaluating the inhibitory effects of UVC at different exposure durations on Candida and Aspergillus. The experimental design is well-structured, data analysis is thorough, and the conclusions hold certain clinical relevance. Overall, this study is innovative and has clinical application prospects, making it suitable for publication after revisions.

Response: We thank the reviewer for their thorough and insightful review of the manuscript, which has greatly contributed to its improvement.

Comment 1. The authors repeatedly mention the limited penetration ability of UVC:
        "Given UVC’s limited depth penetration through a medium [20], it was hypothesized that UVC would not be effective in inhibiting fungal growth."

        "Since UVC light does not penetrate deeply through the cornea [20], combining UVC treatment with antifungal medications supports effectiveness, as indicated by the current study's data."
        "Debriding the corneal epithelium will also improve UVC penetration into the stroma, as the epithelium poses a barrier to UVC light reaching the anterior stroma [20]."

    Does this imply that UVC’s limited penetration is a fundamental technical limitation that is difficult to overcome? Compared to other phototherapy techniques, such as photodynamic therapy (PDT), what advantages does UVC offer?

Response: We appreciate the reviewer’s insightful comment and the opportunity to clarify this point. In our previous studies, we have shown that UVC light has limited depth penetration through the cornea [1]. This limitation has an advantage: it helps protect surrounding structures, such as limbal stem cells, from harm [2]. However, this shallow penetration may also restrict its therapeutic effectiveness in treating deeper infections. That said, corneal infections often involve a disrupted epithelium, and typically, a corneal debridement is performed to collect samples for microbial identification. This procedure is expected to enhance UVC penetration, as the epithelium generally blocks much of the UVC transmission. In contrast, other phototherapy techniques like photodynamic therapy (PDT) are more invasive and require an additional substrate. For example, corneal collagen cross-linking, a PDT procedure, uses UVA and riboflavin to treat keratoconus, a corneal ectatic condition, while also eliminating microorganisms [3]. However, PDT procedures, including corneal collagen cross-linking, are time-consuming and invasive, making them less suitable for routine clinical use. UVC, on the other hand, operates effectively on its own and offers an advantage in this regard. This fact is highlighted in the first paragraph of the introduction (Lines 47-54). The discussion section has also been revised to add this argument (Lines 450-458).

Introduction section Lines 47-54 read as follows: ……………………. Ultraviolet (UV) light has been used safely and effectively in the treatment of a wide range of corneal conditions and ectatic diseases such as keratoconus [8]. In the latter case, UV light (370 nm) is used to excite riboflavin, a chemical substrate, to induce a photodynamic reaction in the procedure known as corneal collagen cross-linking [9, 10]. This method not only treats corneal ectatic diseases but also microbial keratitis through the production of free oxygen radicals [11]. Previous work by our group has demonstrated the efficacy of treatments which use short wavelength (265 nm) UV light alone, classed as UVC, for treating bacterial infections in vitro and in vivo [12]. UVC’s antimicrobial action is attributed to its ready absorbance by microbial DNA, causing cross-linking of adjacent base pairs by photochemical reaction, ultimately resulting in cell death [13, 14]. Therefore, no additional chemical substrate is required to inhibit microbial growth using this wavelength of light.

Discussion section lines 450-458 read as follows………... Given that fungal corneal infections are typically not confirmed until later in their progression, early UVC application as soon as an infiltrate appears may improve treatment outcomes as an empirical approach, inhibiting microbial growth regardless of the infection’s origin. UVC’s limited corneal penetration also reduces off-target damage [31], and its efficacy against fungal pathogens can be enhanced through debridement and combination with antifungal drugs. Additionally, when compared to other phototherapeutic techniques, UVC acts without other agents whereas other photodynamic procedures, e.g. corneal collagen cross-linking require UVA and riboflavin for their antimicrobial effects [9, 10].

Comment 2. Although the study cites previous literature on UVC-induced DNA damage, it does not directly assess the potential harm of UVC to corneal cells in this research.

Response: The aim of this study was to assess the therapeutic potential of UVC against fungal pathogens. However, we have referenced relevant safety assessment studies on the effects of UVC on corneal cells and tissues. Additional safety studies are currently ongoing, independently of the scope of this research. A new reference [31] has also been added to support the argument on UVC safety to corneal limbal cells (See previous paragraph in lines 450-458). Additionally, the safety of the treatment to the cornea has been revised in the discussion section in lines 397-403 as follows.  

Discussion section lines 397-403 read as follows: In previous studies, safety assessments were performed based on 15 s treatment according to bacterial relevance, which suggested that in single application, or in a cumulative dose of 45 s over three days, such UVC exposure was safe in terms of DNA defects in the mouse model tested. These data highlight the likely safety of 30 s UVC but determining the upper safe limit of therapeutic UVC in managing corneal infections would be valuable if repeated exposures are required [19].

Comment 3. The study does not evaluate whether prolonged or repeated use of UVC could lead to sub-inhibitory dose-induced fungal tolerance.

Response: We thank the reviewer for raising this thought-provoking point. Indeed, it represents an intriguing avenue for future research. Ecological evidence shows that certain microorganisms can develop tolerance to UV light and other stressors, though this typically occurs after exposure over several decades. We have previously published a review article [4] discussing the potential for microorganisms to develop resistance to light-based anti-infective technologies. Based on the available evidence, there is currently no strong indication that microorganisms are developing resistance to such technologies. However, given ecological examples, this remains a possibility. Several research groups have recently explored this question, and it would be interesting to observe how fungal species respond to prolonged or repeated sub-lethal doses of UVC. We have added a paragraph to discuss this argument in the discussion section (lines 465-471) with a reference to our review paper.

Discussion section lines 465-471 read as follows: The potential for UVC to induce sub-inhibitory dose-induced fungal tolerance is worthy of future research. While ecological evidence shows that some microorganism-isms can develop tolerance to UV light and other stressors over extended periods, a recent review found limited evidence for tolerance to sub-lethal doses of light-based anti-infective treatments [32]. However, considering ecological examples, it remains a possibility worth exploring, particularly in how fungal species might respond to prolonged or repeated sub-lethal therapeutic UVC doses.

References:

  1. Marasini S, Mugisho OO, Swift S, Read H, Rupenthal ID, Dean SJ, et al. Effect of therapeutic UVC on corneal DNA: Safety assessment for potential keratitis treatment. The Ocular Surface. 2021;20:130-8.
  2. Marasini S, Dean S, Craig JP. Safety of therapeutic ultraviolet C light application at the limbus. Investigative Ophthalmology & Visual Science. 2024;65(7):4121-.
  3. Wollensak G, Spoerl E, Seiler T. Riboflavin/ultraviolet-A–induced collagen crosslinking for the treatment of keratoconus. American journal of ophthalmology. 2003;135(5):620-7.
  4. Marasini S, Leanse LG, Dai T. Can microorganisms develop resistance against light based anti-infective agents? Advanced Drug Delivery Reviews. 2021;175:113822.

Reviewer 2 Report

Comments and Suggestions for Authors

The present study by Bosman et al., entitled “Therapeutic efficacy of Ultraviolet C light on fungal keratitis in vitro and ex vivo studies” focused on exploring therapeutic efficacy of ultraviolet C (UVC) light, alone and along with antifungal drugs to treat microbial keratitis condition. Their results brought forward the antifungal potential of UVC light against fungi Candida albicans and Aspergillus brasiliensis associated with microbial keratitis. Furthermore, the study also demonstrates the synergistic effects of combining UVC with antifungal agents such as natamycin and amphotericin B. Overall, the findings are interesting and relevant in context of treating microbial keratitis condition associated with the pathogenicity C. albicans and A. brasiliensis. The manuscript is well-written and well-executed.

However, there are a few minor comments which the authors need to address:

  1. Line 139-141 in method section: the MIC, …. defined as the lowest dose that inhibited growth, as indicated by a loss of turbidity compared to control,….

How is author measuring MIC? If using CLSI protocol, mention it in method and also provide MIC90 value for both of the antifungals Natamycin and Amphotericin B for both strains, either in a table form or mention it in figure 3 along with each graph.

  1. Figure 5 can be re-arranged in a better way by moving 30s exposure plate to upper panel to make Control, 15s and 30s together and natamycin, 15s+Natamycin and 30s+Natamycin together in lower panel.

  1. In Figure 6, X-axis title is missing…

Author Response

General comment: The present study by Bosman et al., entitled “Therapeutic efficacy of Ultraviolet C light on fungal keratitis in vitro and ex vivo studies” focused on exploring therapeutic efficacy of ultraviolet C (UVC) light, alone and along with antifungal drugs to treat microbial keratitis condition. Their results brought forward the antifungal potential of UVC light against fungi Candida albicans and Aspergillus brasiliensis associated with microbial keratitis. Furthermore, the study also demonstrates the synergistic effects of combining UVC with antifungal agents such as natamycin and amphotericin B. Overall, the findings are interesting and relevant in context of treating microbial keratitis condition associated with the pathogenicity C. albicans and A. brasiliensis. The manuscript is well-written and well-executed.

Response: We thank the reviewer for their thorough and insightful review of the manuscript, which has greatly contributed to its improvement.

Comment 1. However, there are a few minor comments which the authors need to address:

Line 139-141 in method section: the MIC, …. defined as the lowest dose that inhibited growth, as indicated by a loss of turbidity compared to control.

How is author measuring MIC? If using CLSI protocol, mention it in method and also provide MIC90 value for both of the antifungals Natamycin and Amphotericin B for both strains, either in a table form or mention it in figure 3 along with each graph.

Response: The MIC was estimated by using VICTOR Nivo Multimode Plate Reader and measuring optical density. We did not specifically use the CLSI method but used the method indicated in reference [#21] in the manuscript [1]. The method has now been elaborated between lines 133-138 as follows:

Methods section lines 133-138 read as follows: After 24 hours, optical density was measured at 600 nm using VICTOR Nivo Multimode Plate Reader (PerkinElmer, Waltham, MA) to determine the MIC, defined as the lowest dose that inhibited growth. Here we took a reduction in turbidity of at least 90% (i.e. MIC90) to indicate growth inhibition.

Comment 2. Figure 5 can be re-arranged in a better way by moving 30s exposure plate to upper panel to make Control, 15s and 30s together and natamycin, 15s+Natamycin and 30s+Natamycin together in lower panel.

Response: Figure 5 (now figure 6) has been rearranged as suggested.

Comment 3. In Figure 6, X-axis title is missing…

Response: X-axis title (now figure 7, treatment exposures and units to UVC exposures) has been revised now.

Reference:

[1]. Andrews JM. Determination of minimum inhibitory concentrations. Journal of antimicrobial Chemotherapy. 2001;48(suppl_1):5-16.

Reviewer 3 Report

Comments and Suggestions for Authors

The study by Bosman M et al, explores UVC light as an antifungal treatment for corneal infections, showing effective fungal inhibition alone and enhanced efficacy when combined with antifungal drugs. I have no major issues with the study apart from the following minor critique that should improve the overall quality of the manuscript.

Abstract:

  • Line 16: Specify the concentration of the antifungal drugs used in the combined treatment.
  • Line 20: ItalicizeCandida albicans.
  • Line 22: In the first instance, write outCandida albicans and Aspergillus brasiliensis in full. After that, you may refer to them as  albicans and A. brasiliensis respectively throughout the manuscript. Ensure that both Candida and Aspergillus are followed by their species names throughout the manuscript.

Methods:

  • Line 84:The phrase "2-3 colony-forming units (CFU)" should be revised to "2–3 colonies" for clarity and accuracy.
  • Lines 90–95:While the described technique is common, it would be beneficial to specify whether the spores were filtered to remove hyphal fragments, ensuring a uniform inoculum. If no filtration was performed, this should be explicitly stated.
  • Section 2.3 (In Vitro Fungal Keratitis Model):The authors state that 100 μL of  albicans (10⁷ cells) or A. brasiliensis (10⁶ spores) was added to 1 mL of molten YPD top agar (0.75% w/v agar in YPD). Does this mean C. albicans and A. brasiliensis were added directly to heated molten agar? Further, it is mentioned that three 5 μL aliquots (containing 10⁵ Candida cells or 10⁴ Aspergillus spores) were pipetted into the center of each well of a 96-well plate. The methodology should be revised for clarity and precision.
  • Line 108: It seems that experiments were performed in a 96-well flat-bottom microplate; however, Figures 1B and 4B appear to depict petri plates. Please clarify this.
  • Line 134:Table 1 appears redundant, as the same information is presented in Figure 3. Consider removing the table for conciseness.
  • Line 148:The phrase "Ater one hour" should be corrected to "After one hour."
  • Section 2.6:The study states that corneas were stored at 4°C for up to one week. Were viability markers (e.g., epithelial integrity, metabolic activity) assessed to confirm that stored corneas remained representative of live tissue? A brief mention of corneal integrity validation would strengthen the reliability of the model.

Results:

  • Provide the concentration of the antifungal drugs used in the combined treatment. It appears that the antifungals were used at their minimum inhibitory concentrations (MICs) as stated in line 16. If the treatment was performed at the MIC values, it could potentially kill the fungus on its own. Please clarify this point.
  • To make the results more robust, it would be beneficial to detect live and dead fungal cells under all conditions using additional methods, such as the Live/Dead Cell Viability Assay, ATP Bioluminescence Assay, or XTT Reduction Assay.
  • Line 218: two time “in” repeated.

Figure Legends:

  • For Figure 4, include the number of biological replicates, and clarify which statistical methods were used to determine significance.
  • The Figure 5 legend is missing information about the number of replicates and biological repeats. Please include these details in the figure legend.

Discussion:

The phrase “Candida and Aspergillus species are among the predominant species implicated in MK” (Line 276) should specify which species were tested in this study to avoid generalization.

Author Response

General comment: The study by Bosman M et al, explores UVC light as an antifungal treatment for corneal infections, showing effective fungal inhibition alone and enhanced efficacy when combined with antifungal drugs. I have no major issues with the study apart from the following minor critique that should improve the overall quality of the manuscript.

Response: We thank the reviewer for their thorough review of the manuscript and encouraging feedback.

Comment 1. Abstract: Line 16: Specify the concentration of the antifungal drugs used in the combined treatment.

Response: The concentrations have now been specified in abstract lines 23-25 in the abstract and methods section (lines 141-143) as follows:

Abstract, lines 23-25: The broth MICs were 0.1875 µg/mL for Amphotericin B against C. albicans, 6.25 µg/mL against A. brasiliensis, and 0.78125 µg/mL for Natamycin against C. albicans, 7.8125 µg/mL against A. brasiliensis.

Comment 2. Line 20: Italicize Candida albicans.

Response: This has been corrected now.

Comment 3. Line 22: In the first instance, write out Candida albicans and Aspergillus brasiliensis in full. After that, you may refer to them as albicans and A. brasiliensis respectively throughout the manuscript. Ensure that both Candida and Aspergillus are followed by their species names throughout the manuscript.

Response: The names of microorganisms have been written out fully the first time and abbreviated thereafter throughout the manuscript. In instances where the notation is used alongside drugs, it has been retained for convenience (e.g., Figure 4, lines 264-269).

Comment 4. Methods: Line 84: The phrase "2-3 colony-forming units (CFU)" should be revised to "2–3 colonies" for clarity and accuracy.

Response: This has now been revised accordingly (Line 77). 

Comment 5. Lines 90–95: While the described technique is common, it would be beneficial to specify whether the spores were filtered to remove hyphal fragments, ensuring a uniform inoculum. If no filtration was performed, this should be explicitly stated.

Response: Spores were not filtered, and this point has been stated now as follows:

Line 85: ‘……….agar surface but without filtering into 10 mL sterile saline…….’

Comment 6. Section 2.3 (In Vitro Fungal Keratitis Model): The authors state that 100 μL of  albicans (10⁷ cells) or A. brasiliensis (10⁶ spores) was added to 1 mL of molten YPD top agar (0.75% w/v agar in YPD). Does this mean C. albicans and A. brasiliensis were added directly to heated molten agar? Further, it is mentioned that three 5 μL aliquots (containing 10⁵ Candida cells or 10⁴ Aspergillus spores) were pipetted into the center of each well of a 96-well plate. The methodology should be revised for clarity and precision.

Response: The paragraph has been revised for clarity and precision. The new text reads as follows:

Lines 98-106: An in vitro infection model simulating subsurface corneal infections was adapted from previous work using Pseudomonas aeruginosa [12]. In brief, 100 µl of either C. albicans (107 cells) or A. brasiliensis (106 spores) was mixed with 1 mL of warm molten YPD top agar (0.75% w/v agar in YPD) and aliquoted into a 96-well flat-bottom microplate. Three 5 µL (containing 105 cells of C. albicans or 104 spores of A. brasiliensis) were added to the centre of each well. Once solidified, the droplets formed a circular subsurface fungal corneal infection model, which was used to assess the therapeutic potential of UVC, both alone and combined with conventional antifungal drugs, in three independent experiments. The experimental protocol has been represented graphically in Figure 1.

Comment 7. Line 108: It seems that experiments were performed in a 96-well flat-bottom microplate; however, Figures 1B and 4B appear to depict petri plates. Please clarify this.

Response: We appreciate the reviewer’s comment regarding the ambiguity in the chronology of the methods. The therapy testing was indeed conducted in a 96-well flat-bottom microplate. After UVC exposure, the in vitro infection model described in section 2.3 was dissolved in saline, diluted, and plated. Figures 1B and 4B show the colonies recovered after dilution and plating. To address the current ambiguity, we have added a schematic diagram (New Figure 1) to further clarify the methods used in these experiments. Subsequent figure numbers have now changed accordingly.

Comment 8. Line 134: Table 1 appears redundant, as the same information is presented in Figure 3. Consider removing the table for conciseness.

Response: The table has been removed as suggested.

Comment 9. Line 148: The phrase "Ater one hour" should be corrected to "After one hour."

Response: This typo has been corrected in what is now line 148. Thank you.

Comment 10. Section 2.6: The study states that corneas were stored at 4°C for up to one week. Were viability markers (e.g., epithelial integrity, metabolic activity) assessed to confirm that stored corneas remained representative of live tissue? A brief mention of corneal integrity validation would strengthen the reliability of the model.

Response: Corneal integrity was assessed based on structural epithelial damage only. The goal of these experiments was to test the in vitro findings using an ex vivo tissue model for potential translation. Obtaining ex vivo eyes from a slaughterhouse and keeping the tissue alive is challenging, and eyes lack active immunity. Beyond visually assessing tissue integrity, viability markers were not tested, as it did not impact the evaluation of the current experimental outcomes. Building from this ex vivo data, we are now undertaking testing of the therapy in a live animal model, which will take into consideration all the factors mentioned in the comment above. A statement has been added at lines 157-160 to clarify as follows:

Lines 157-160: As a step towards future testing for potential clinical translation in an in vivo model with active immunity, the goal of the current experiments was to explore the in vitro findings using an ex vivo porcine keratitis model Corneal integrity was assessed for visually to confirm the epithelium was intact, and any eyes with epithelial damage were excluded.

Comment 11. Results: Provide the concentration of the antifungal drugs used in the combined treatment. It appears that the antifungals were used at their minimum inhibitory concentrations (MICs) as stated in line 16. If the treatment was performed at the MIC values, it could potentially kill the fungus on its own. Please clarify this point.

Response: The concentrations of the antifungal drugs used in the combined treatment are reported in lines 142-145. The MICs were determined in YPD broth following a standard protocol [1]. However, the combined treatment (UVC + antifungal drugs at MIC) was tested in a semi-solid agar droplet, where the antifungal drugs at their MIC values would not independently kill the fungus due to the different experimental conditions (liquid vs semi-solid experimental setup). This has been clarified in lines 120-122 as follows:

Lines 120-122: The MIC determined in YPD broth conditions (liquid medium) was used to establish the objective drug dosage for the combination therapy in the semi-solid in vitro infection model described above.

Comment 12. To make the results more robust, it would be beneficial to detect live and dead fungal cells under all conditions using additional methods, such as the Live/Dead Cell Viability Assay, ATP Bioluminescence Assay, or XTT Reduction Assay.

Response: We thank the reviewer for this useful feedback. These were independently conducted proof of concept studies to test UVC efficacy in both in vitro and ex vivo infection models and are reported as such. Our ongoing experiments have been incorporating the Live/Dead Cell Viability Assay in biofilm conditions, along with other methods, to assess the robustness of efficacy testing for these pathogens, and we will also consider the additional suggested methods mentioned in our future experiments.

Comment 13. Line 218: two time “in” repeated.

Response: This has now been corrected.

Comment 14. Figure Legends: For Figure 4, include the number of biological replicates, and clarify which statistical methods were used to determine significance.

Response: Thank you for this comment. All the figure legends have now been revised.

Comment 15. The Figure 5 legend is missing information about the number of replicates and biological repeats. Please include these details in the figure legend.

Response: Thank you for this comment. All the figure legends have now been revised.

Comment 16. Discussion: The phrase “Candida and Aspergillus species are among the predominant species implicated in MK” (Line 276) should specify which species were tested in this study to avoid generalization.

Response: The species used in the present study have now been stated in the following sentence. The revised text reads as follows:

Lines 374-377: Candida and Aspergillus species are among the predominant species implicated in MK [23, 24] and represent a strong starting point for understanding the efficacy of this therapy in the realm of fungal keratitis. We tested C. albicans and A. brasiliensis as model pathogens in the present study. The data presented here strongly indicate an inhibitory effect of UVC exposure on both the fungal species tested in an in vitro fungal corneal infection model.

Reference:

[1]. Andrews JM. Determination of minimum inhibitory concentrations. Journal of antimicrobial Chemotherapy. 2001;48(suppl_1):5-16.

Reviewer 4 Report

Comments and Suggestions for Authors

The research article titled "Therapeutic efficacy of Ultraviolet C light on fungal keratitis— in vitro and ex vivo studies" is recommended for publication in Antibiotics with major revisions.

This study investigates the antifungal efficacy of ultraviolet C light (UV-C) against Candida albicans and Aspergillus brasiliensis, two major fungal pathogens causing keratitis. Using both in vitro and ex vivo models, the authors evaluate UV-C treatment alone and in combination with antifungal drugs. The results demonstrate that short-term UV-C exposure significantly impairs fungal growth, highlighting its potential as an alternative therapeutic strategy for corneal keratitis. The research aligns well with the scope of Antibiotics, particularly in exploring antifungal agents and combination therapies for managing corneal keratitis. These findings provide valuable insights for researchers and clinicians in the fields of antimicrobial drug development and ophthalmic infections. However, the Methods section requires a more precise and detailed description to enhance data reproducibility, and additional explanation of Aspergillus under UV-C treatment is recommended. Based on these considerations, I recommend this manuscript for publication after major revisions. Below are specific comments for the authors to address:

  1. Experimental Design Clarification: The methodology and experiments in this study are well-executed; however, additional details are needed to improve clarity and ensure data reproducibility.

(1) A schematic diagram or photograph is recommended to illustrate the experimental setup of the UV-C light equipment, particularly for both subsurface and liquid broth experiments, to enhance reader understanding.

(2) Natamycin is known for its poor water solubility, as I have previously experienced. Therefore, the solvent and preparation methods for natamycin and amphotericin B in the MIC tests should be clearly described in the Methods section.

(3) The contents of Table 1 should be reconsidered. Since Figure 3 already presents the MIC results of both antifungal drugs, Table 1 may be misleading, as readers might expect it to display MIC test results rather than reference data.

(4) In Section 2.4, details regarding the dilution buffer and incubation conditions (e.g., temperature, shaking) for both fungi are unclear and should be specified. Additionally, in Section 2.5.2, the equipment used for OD600 measurements should be mentioned.

(5) In Section 2.5.2, natamycin was applied at 10 µL, while amphotericin B was applied at 100 µL. The rationale for this volume difference should be clarified.

  1. Results of UV-C Treatment on Aspergillus in Vitro (Lines 200–205): Further explanation is needed for this set of data. As expected from the presented data, fungal reduction correlates positively with exposure time. However, the CFU reduction at 30 seconds appears significantly greater than at 60 seconds, which is unexpected. The authors should clarify this anomaly and provide possible explanations.
  2. Discussion on Pathogen and Drug Selection: The manuscript provides valuable background information on microbial keratitis and UV-C treatment. However, further discussion is needed on the rationale behind selecting specific pathogens and antifungal agents. Both natamycin and amphotericin B belong to the polyene macrolide class, whereas azoles, such as voriconazole, are another major class of antifungal agents. To comprehensively evaluate UV-C treatment in this combination study, it is important to include agents from different classes. The authors should elaborate on why azoles were not considered in this study. Additionally, a previously published study by the authors’ group explored UV-C’s antifungal potential against a broader range of microorganisms. Discussing how the current study builds upon that work and highlighting its key advancements would strengthen the manuscript’s impact and innovation.
  3. Minor Revisions:

(1) Line 20 (Abstract): C. albicans should be italicized.

(2) Line 55: Typo—“radicle” should be corrected.

(3) Line 148: Typo—“Ater” needs correction.

(4) Line 149: Redundant words should be removed.

(5) Figure 1: Missing annotation for ‘*’; additionally, the meanings of ‘N, 1, 2, 3’ in the images should be explained in the figure legend.

(6) The spacing between numbers and the unit “seconds (s)” should be consistent throughout the manuscript.

Author Response

General comment: The research article titled "Therapeutic efficacy of Ultraviolet C light on fungal keratitis— in vitro and ex vivo studies" is recommended for publication in Antibiotics with major revisions.

This study investigates the antifungal efficacy of ultraviolet C light (UV-C) against Candida albicans and Aspergillus brasiliensis, two major fungal pathogens causing keratitis. Using both in vitro and ex vivo models, the authors evaluate UV-C treatment alone and in combination with antifungal drugs. The results demonstrate that short-term UV-C exposure significantly impairs fungal growth, highlighting its potential as an alternative therapeutic strategy for corneal keratitis. The research aligns well with the scope of Antibiotics, particularly in exploring antifungal agents and combination therapies for managing corneal keratitis. These findings provide valuable insights for researchers and clinicians in the fields of antimicrobial drug development and ophthalmic infections. However, the Methods section requires a more precise and detailed description to enhance data reproducibility, and additional explanation of Aspergillus under UV-C treatment is recommended. Based on these considerations, I recommend this manuscript for publication after major revisions. Below are specific comments for the authors to address:

Response: We thank the reviewer for their time and expertise in evaluating our manuscript and helping to improve it.

Comment 1.

Experimental Design Clarification: The methodology and experiments in this study are well-executed; however, additional details are needed to improve clarity and ensure data reproducibility.

    • A schematic diagram or photograph is recommended to illustrate the experimental setup of the UV-C light equipment, particularly for both subsurface and liquid broth experiments, to enhance reader understanding.

Response: Thank you for this suggestion. We have now included the experimental protocol in graphical form – Figure 1. 

Comment 2. Natamycin is known for its poor water solubility, as I have previously experienced. Therefore, the solvent and preparation methods for natamycin and amphotericin B in the MIC tests should be clearly described in the Methods section.

Response: Both natamycin and amphotericin were dissolved in DMSO. Natamycin was prepared at a stock concentration of 1 mg/mL and amphotericin B at 5 mg/mL in DMSO. These stock solutions were stored and subsequently diluted to working concentrations using serial dilution as needed. This information has been added in lines 124-127. The revised text reads as follows:

Lines 124-127: To determine MICs, both natamycin and amphotericin B were dissolved in DMSO. Natamycin was prepared at a stock concentration of 1 mg/mL and amphotericin B at 5 mg/mL in DMSO. These stock solutions were stored and subsequently diluted to working concentrations using serial dilution as needed.

Comment 3. The contents of Table 1 should be reconsidered. Since Figure 3 already presents the MIC results of both antifungal drugs, Table 1 may be misleading, as readers might expect it to display MIC test results rather than reference data.

Response: This point has been addressed by removing Table 1 as also suggested by Reviewer 3.

Comment 4. In Section 2.4, details regarding the dilution buffer and incubation conditions (e.g., temperature, shaking) for both fungi are unclear and should be specified. Additionally, in Section 2.5.2, the equipment used for OD600 measurements should be mentioned.

Response: These details have been added as suggested. The new text reads as follows:

Lines 133-138: C. albicans was incubated at 37 °C at 200 rotations per minute (rpm), while A. brasiliensis was incubated at 26 °C, with shaking. After 24 hours, optical density was measured at 600 nm using VICTOR Nivo Multimode Plate Reader (PerkinElmer, Waltham, MA) to deter-mine the MIC, defined as the lowest dose that inhibited growth. A reduction in turbidity of at least 90% (i.e. MIC90) was taken to indicate growth inhibition.

Comment 5. In Section 2.5.2, natamycin was applied at 10 µL, while amphotericin B was applied at 100 µL. The rationale for this volume difference should be clarified.

Response: Initially, we applied 10 µL of both natamycin and amphotericin B. While natamycin demonstrated significant growth inhibition, amphotericin B did not produce similar results. Both drugs were intended to be compared as treatments were expected to show higher efficacy than the untreated control. However, with 10 µL of amphotericin B, we did not observe enhanced growth inhibition against Aspergillus in the in vitro infection model, which led us to explore higher volumes. Natamycin showed effectiveness even at smaller volumes, while amphotericin B required larger volumes to achieve comparable efficacy during optimization. Consequently, we adjusted the volumes for each drug to optimize their effects. This argument has now been added in lines 149-155.

Lines 149-155: In preliminary tests evaluating the antifungal efficacy of natamycin and amphotericin B at their respective MICs using the in vitro semisolid infection model, 10 µL of each drug was applied to assess treatment effectiveness compared to an untreated control. The results indicated that while 10 µL of natamycin demonstrated the desired partial efficacy in inhibiting fungal growth at its MIC, a higher volume of amphotericin B was required to achieve similar effects. Consequently, the antifungal volumes were adjusted accordingly for each drug.

Comment 6. Results of UV-C Treatment on Aspergillus in Vitro (Lines 200–205): Further explanation is needed for this set of data. As expected from the presented data, fungal reduction correlates positively with exposure time. However, the CFU reduction at 30 seconds appears significantly greater than at 60 seconds, which is unexpected. The authors should clarify this anomaly and provide possible explanations.

Response: Our data showed that when comparing CFU counts within the UVC treatment groups of 15 seconds and longer, the inhibition rates were similar (ANOVA, p = 0.3811), despite apparent visual differences. Additionally, a comparison between the 30s and 60s groups revealed no significant difference (t-test, p = 0.2515). This observation is addressed in the results section. Together, these findings suggest that once the fungal inactivation threshold is reached, increasing the UVC dose further does not enhance the killing efficacy in this experimental setup.

Comment 7. Discussion on Pathogen and Drug Selection: The manuscript provides valuable background information on microbial keratitis and UV-C treatment. However, further discussion is needed on the rationale behind selecting specific pathogens and antifungal agents.

Response: We selected Candida albicans and Aspergillus brasiliensis due to their relative abundance in microbial keratitis [1, 2], including in cases of contact lens wear. Both organisms were also readily available for testing in the laboratory. This information has been revised in lines 375-377 as follows:

Lines 375-377: Candida and Aspergillus species are among the predominant species implicated in MK [23, 24] and represent a strong starting point for understanding the efficacy of this therapy in the realm of fungal keratitis. We tested C. albicans and A. brasiliensis as model pathogens in the present study.

Comment 8. Both natamycin and amphotericin B belong to the polyene macrolide class, whereas azoles, such as voriconazole, are another major class of antifungal agents. To comprehensively evaluate UV-C treatment in this combination study, it is important to include agents from different classes. The authors should elaborate on why azoles were not considered in this study.

Response: We thank the reviewer for this insightful comment and the opportunity to provide further clarification. We initially planned to include voriconazole and fluconazole in the study, but issues regarding the safe disposal of these drugs delayed our ability to include them. Reviewing clinical use for MK, natamycin and amphotericin B are the most commonly used drugs and voriconazole and fluconazole are not generally used as first-line drugs [2, 4].  We therefore proceeded with natamycin and amphotericin B.

Comment 9. Additionally, a previously published study by the authors’ group explored UV-C’s antifungal potential against a broader range of microorganisms. Discussing how the current study builds upon that work and highlighting its key advancements would strengthen the manuscript’s impact and innovation.

Response: Thank you for this important comment. In our previous research, we tested a broad range of microorganisms using agar plates as surface lawns, without considering depth factors, except for Pseudomonas aeruginosa. In the current experiments, we specifically examined the depth factor by utilizing an in vitro infection model and an ex vivo model to study infection in subcellular contexts. This new data builds upon our previous in vitro studies and lays the groundwork for transitioning to in vivo research. This discussion has now been added to the manuscript as follows:

Lines 459-464: In our previous research, we tested a broad range of microorganisms using agar plates as surface lawns, without considering depth factors, except for Pseudomonas aeruginosa. In the current experiments, we specifically examined the depth factor by utilizing an in vitro infection model and an ex vivo model to study infection in subcellular contexts for fungal keratitis. This new data builds upon our previous in vitro studies and lays the groundwork for transitioning to in vivo research.

Comment 10. Minor Revisions:

  • Line 20 (Abstract): albicans should be italicized.

Response: This has been corrected throughout the manuscript. Thank you.

  • Line 55:Typo—“radicle” should be corrected.

Response: radicle has been changed to radicals. Thank you. Line 48.

  • Line 148:Typo—“Ater” needs correction.

Response: This has been corrected. Thank you. Line 148.

  • Line 149:Redundant words should be removed.

Response: This has been corrected. Thank you.

  • Figure 1:Missing annotation for ‘*’; additionally, the meanings of ‘N, 1, 2, 3’ in the images should be explained in the figure legend.

Response: This has been added. Now figure 2, line 215-221.

(6) The spacing between numbers and the unit “seconds (s)” should be consistent throughout the manuscript.

Response: This has been corrected throughout the manuscript.

References:

[1]. Tanure MAG, Cohen EJ, Sudesh S, Rapuano CJ, Laibson PR. Spectrum of Fungal Keratitis at Wills Eye Hospital, Philadelphia, Pennsylvania. Cornea. 2000;19(3):307-12.

[2]. Liesegang TJ, Forster RK. Spectrum of microbial keratitis in South Florida. American journal of ophthalmology. 1980;90(1):38-47.

[3.] Iyer SA, Tuli SS, Wagoner RC. Fungal keratitis: emerging trends and treatment outcomes. Eye & contact lens. 2006;32(6):267-71.

[4]. Bunya VY, Hammersmith KM, Rapuano CJ, Ayres BD, Cohen EJ. Topical and oral voriconazole in the treatment of fungal keratitis. American journal of ophthalmology. 2007;143(1):151-3.

Round 2

Reviewer 4 Report

Comments and Suggestions for Authors

The manuscript is now well-refined and is recommended for acceptance with one minor revision. The authors’ efforts in addressing all comments and suggestions from the previous review are sincerely appreciated. The revisions have been implemented with clarity and precision, significantly enhancing the quality and comprehensibility of the manuscript.

The additional explanations of the experimental methods substantially improve the study’s reproducibility and scientific rigor. Moreover, the authors have provided an excellent discussion that highlights the advancements of this study compared to their previous research, further strengthening the impact of their work. Herein, all concerns raised in the previous review report have been thoroughly addressed.

However, the new Figure 4 still requires minor improvements to ensure consistency with the other figures. The resolution and font clarity should be enhanced for better readability.

Once again, I sincerely appreciate the authors’ diligence. This study presents innovative models for antifungal treatment using UV-C, and its findings will be highly valuable to researchers in the field of antimicrobial drug development.